# Enhancement of Antibacterial Performance of Silver Nanowire Transparent Film by Post-Heat Treatment

**DOI:** 10.3390/nano10050938

**Published:** 2020-05-13

**Authors:** Ji-Hyeon Kim, Junfei Ma, Sungjin Jo, Seunghun Lee, Chang Su Kim

**Affiliations:** 1Advanced Nano-Surface Department, Korea Institute of Materials Science, Changwon 51508, Korea; sangdu87@kims.re.kr (J.-H.K.); ready@kims.re.kr (J.M.); seunghun@kims.re.kr (S.L.); 2School of Architectural, Civil, Environmental, and Energy Engineering, Kyungpook National University, Daegu 41566, Korea; sungjin@knu.ac.kr

**Keywords:** silver nanowire, antibacterial activity, heat treatment, Rayleigh–Plateau instability, oxidation

## Abstract

Silver nanomaterials (AgNMs) have been applied as antibacterial agents to combat bacterial infections that can cause disease and death. The antibacterial activity of AgNMs can be improved by increasing the specific surface area, so significant efforts have been devoted to developing various bottom-up synthesis methods to control the size and shape of the particles. Herein, we report on a facile heat-treatment method that can improve the antibacterial activity of transparent silver nanowire (AgNW) films in a size-controllable, top-down manner. AgNW films were fabricated via spin-coating and were then heated at different temperatures (230 and 280 °C) for 30 min. The morphology and the degree of oxidation of the as-fabricated AgNW film were remarkably sensitive to the heat-treatment temperature, while the transparency was insensitive. As the heat-treatment temperature increased, the AgNWs spontaneously broke into more discrete wires and droplets, and oxidation proceeded faster. The increase in the heat-treatment temperature further increased the antibacterial activity of the AgNW film, and the heat treatment at 280 °C improved the antibacterial activity from 31.7% to 94.7% for *Staphylococcus aureus,* and from 57.0% to 98.7% for *Escherichia coli*. Following commonly accepted antibacterial mechanisms of AgNMs, we present a correlation between the antibacterial activity and surface observations of the AgNW film.

## 1. Introduction

Nowadays, public awareness of environmental and health issues is increasing. Of the issues, bacterial infections and the resulting diseases and deaths are recognized as a significant danger to public health worldwide [1,2,3,4,5]. Accordingly, disinfectants such as antibacterial agents and antibiotics have been developed [6]. Silver nanomaterials (AgNMs) as antibacterial agents have a wide range of potential applications in the medical, scientific, food, and pharmaceutical industries, thanks to their effective antibacterial activity against Gram-positive bacteria and Gram-negative bacteria [7] and their low toxicity to human cells at a low concentration [8]. Nanomaterials are commonly defined as materials which are 1–100 nm in size in at least one dimension [9]. Many synthesis methods for various forms (i.e., nanowires, nanocubes, nanocoils, nanorods, and nanoparticles) of AgNMs have been developed so far [10,11,12,13]. These AgNMs have shown much better broad-spectrum antibacterial efficiency than the bulk silver counterpart [14]. 

Of AgNMs, silver nanowire (AgNW) has a unique structure, an extremely long structure along the [110] direction with a diameter of 10–200 nm, a length of 5–100 μm, and an aspect greater than 10 [15]. As an antibacterial agent, AgNW possesses an inherent structural advantage that enables researchers to maximize the great specific surface area and so the surface-dependent antibacterial activity thanks to the anti-overlapping characteristic when in film form. AgNW film in a mesh form can be fabricated by various methods, such as spin coating, drop casting, dip coating, spray coating, rod-coating, vacuum filtration, and slot-die coating. This mesh-shaped film structure provides transparency (i.e., an optical advantage) to the film. In the electrode field, AgNW has been most frequently used as transparent electrode film source [16]. Moreover, with regard to orientation, AgNW dominantly consisting of {100} facets of the sidewall showed more powerful antibacterial performance than Ag spherical nanoparticles (NPs) mainly composed of {111} facets [17]. This is because a {100} facet with a high surface energy and so a high reactivity was rapidly attached into bacteria membrane with a large contact surface area, compared to a {111} facet [17]. Accordingly, AgNW has exhibited a strong antibacterial activity against at least 12 kinds of bacteria, including *Escherichia coli* (*E. coli*), *Staphylococcus aureus* (*S. aureus*), and several antibiotic-resistant bacteria (e.g., *Methicillin-resistant Staphylococcus aureus* and *Pseudomonas aeruginosa*) [18,19,20].

However, Visnapuu et al. [21] and Helmlinger et al. [22] experimentally revealed that AgNWs possess lower antibacterial activity than other AgNMs in the form of nanospheres, nanocubes, and nonoplates. They suggested the lower antibacterial activity of AgNWs is due to the smaller specific surface area and hence slower release of positively-charged silver ions (Ag^+^) [21,22]. It is widely accepted that Ag^+^ release is a main source for the antibacterial actions of AgNMs [21,22,23,24,25]. 

Thus, to further improve the antibacterial performance of AgNWs, the application of downsizing treatment can be a good strategy. At high temperatures above 200 °C, AgNW has shown structural decomposition into discrete short wires and droplets due to the Rayleigh–Plateau instability [26,27,28,29,30,31]. This breakup phenomenon has been studied as a structural failure by joule heating of AgNW electrodes [26,27,28,29,30,31]. As the applied temperature of AgNW increased by joule heating or its simulated heat-treatment, the AgNWs were broken into shorter wires and smaller particles [29,30,32,33,34]. Therefore, for AgNWs as an antibacterial material, heat-treatment can be considered as a top-down synthesis method to downsize AgNWs. 

This study investigates heat treatment as a potential technique to improve the functional performance of AgNW antibacterial films. The AgNW films fabricated by spin-coating method were heated at different temperatures. As a function of heat-treatment temperature, the morphology, chemical state, optical properties, and sheet resistance were inspected by using field emission scanning electron microscopy (FE-SEM), ultraviolet–visible spectroscopy (UV-Vis), X-ray photoelectron spectroscopy (XPS), and four-probe. Additionally, the antibacterial test against *E. coli* was carried out. By comparative analysis, we discuss how the heat-treatment temperature affects the antibacterial activity of AgNW film on the basis of the inspected characteristics.

## 2. Materials and Methods 

### 2.1. Material, Fabrication, and Heat Treatment 

AgNW dispersion solution (C3NANO corp., Hayward, CA, USA), where AgNWs of 0.3 wt % are dispersed in ethanol, was used in this study. Each AgNW with a diameter of 20–30 nm and a length greater than 20 μm was capped with a 2–3 nm thick polyvinylpyrrolidone (PVP) layer, preventing aggregation. Using the AgNW solution, AgNW films were fabricated by the spin-coating technique at 7000 rotations per minute for 30 s, and then sufficiently dried at room temperature. The as-fabricated AgNW films were heated in an air-oven for 30 min at 230 and 280 °C, respectively. Hereafter, the as-fabricated AgNW film and the AgNW films heated at 230 and 280 °C are denoted as “AgNW”, “AgNW-230 °C”, and “AgNW-280 °C”, respectively.

The heating temperatures (230 and 280 °C) were selected with respect to the size and shape of the heated AgNWs. Theoretically, the greater the specific surface area of antibacterial substances (i.e., AgNW), the higher the antibacterial activity. The heating temperatures of 200–300 °C (at 10 °C intervals) were assessed by SEM image analysis. With increasing the heating temperature, the AgNWs were continuously cut into more segments. The average lengths of the AgNWs heated at 200, 230, 250, and 280 °C were 8.3 (±3.6), 2.5 (±1.0), 0.5 (±0.3), and 0.4 (±0.3) nm, respectively. On the other hand, the AgNWs heated at 200 °C showed the partial thickening without notable segmentation, and the Ag segments were notably thickened from 280 °C. Given the specific surface area, the thickening is inappropriate. In addition, it was observed that generated Ag_2_O particles were dispersed on the glass substrate from 250 °C. The AgNW films heated at 230 and 280 °C were outstandingly different in the segment form, and also seemed suitable in terms of the specific surface area and antibacterial activity. 

### 2.2. Characterization 

The surface topologies of the three samples were examined by using FE-SEM (JSM-7610F, JEOL co., Tokyo, Japan) operating at 5 kV voltage and 10 μA current under a high vacuum condition. The thickness was measured by Alpha-step (DektakXT, BRUKER co., Billerica, MA, USA). The chemical qualification and quantification were performed by XPS (K-Alpha, Thermo Fisher co., Waltham, MA, USA) of which the X-ray source was an Al K Alpha gun. The spot size and energy step size of the XPS analysis were 400 μm and 1.0 eV, respectively. The optical properties were also evaluated; the transmittance was investigated by UV-Vis (Cary 5000 UV-Vis-NIR, Agilent co., Santa Clara, CA, USA), while total light transmittance and haze were measured by using a haze meter (COH 400, Nippon Denshoku co., Tokyo, Japan). Haze means a degree of light scattering that is defined as the percentage of the light diffusely scattered more than 2.5° from the incident light, compared to the total transmitted light. The sheet resistance was investigated by using a four-point probe. 

### 2.3. Antibacterial Test 

The antibacterial activity of the AgNW films were estimated according to the Japanese Industrial Standard (JIS) Z 2801 [35]. We performed antibacterial tests against *E. coli* (Korea Collection for Type Cultures (KCTC), Jeongeup, Korea) and *S. aureus* (KCTC, Jeongeup, Korea), which are representative Gram-negative bacteria and Gram-positive bacteria, respectively. The inoculum liquid containing the test strain was prepared via the following process: the test strain was grown in 1/500 nutrient broth (i.e., a liquid culture) overnight, and then the liquid was diluted with Soybean Casein Digest Lecithin Polysorbate (SCDLP) broth liquid to reach a density of 10^7^ colony forming units per milliliter (CFU/mL), the greatest density regulated in the JIS Z 2801 [35]. Then, 0.4 mL of the as-prepared inoculum liquid was injected onto 40 mm × 40 mm test area of the test samples (i.e., AgNW, AgNW-230 °C, and AgNW-280 °C) and control (standard film). To achieve the reliability of the antibacterial test, all the samples were tested in triplicate. The test area was covered with a sterile cover film and transferred into a sterile petri dish. The petri dish was incubated at 35 (± 1) °C for 30 min in a humid environment. This 30 min corresponds to the contact time between the test strain and test plane. After the incubation, the bacteria attached on the test area and cover film were sufficiently extracted into 10 mL of SCDLP broth. One milliliter of the extraction was diluted by mixing with 9 mL of phosphate-buffered saline. This dilution was consecutively repeated two (samples) or three (control) times more. The viable bacteria number in 1 mL of the final diluted liquid was obtained by using the agar plate count method. From the average number of viable bacteria and the dilution factor, the antibacterial activity in percentage reduction and log reduction was calculated with reference to the control, according to the JIS Z 2801 [35].

## 3. Results and Discussion

### 3.1. Surface Morphology

As shown in Figure 1a, the AgNW mesh films were fabricated on glass by the spin coating method. The thickness measured by Alpha-step was 54 ± 6 nm. The substrate coverage was 26.5% ± 5.9%, which was identified by analyzing the five SEM images of 2000 magnification of the bare AgNW film with the “ImageJ” program. As can be identified in Figure 1, the AgNWs broke into shorter wires and more large-droplets (i.e., more segmentations) with increasing the applied heating temperature. In addition, a part of the AgNWs broken by the heat treatment showed thickening at the end or all parts. These morphological changes have been also reported by Oh et al. [34] and Wang et al. [36]. In the case of the AgNW-280 °C (Figure 1c), discrete Ag_2_O NPs less than 50 nm in size were observed (this is elucidated in detail in the next subsection). The AgNW film heated at 200 °C (not contained in this study) showed the thickening of the AgNWs rather than the fragmentation, while the AgNW film heated at 250 °C (not contained in this study) showed a similar morphology to the AgNW-280 °C. The fragmentation and thickening of AgNWs are guessed as being spontaneous processes to minimize the surface energy and hence the total system energy through self-coalescence. It is believed that the fragmentation is driven by the Rayleigh–Plateau instability in which a fluid spontaneously breaks up into smaller droplets with larger surface-to-volume ratios to reduce the total system energy [31,37,38]. In particular, nanostructured materials including AgNW are more likely to cause self-coalescence owing to the high specific surface area, high surface energy, and hence low thermodynamic stability [39]. 

High environmental temperatures (normally, above 200 °C) cause AgNWs to fuse, and concurrently provide an energy for the spontaneous self-coalescence by surface diffusion [29,30,32,33,34,37,40]. The excessive self-coalescence causes the fragmentation of AgNWs. Aveek et al. [31] experimentally proved that the diameter of AgNWs influenced the onset temperature of the fragmentation: thinner nanowires broke at lower temperatures. Mayoral [41] recorded the in situ fragmentation of a AgNW at 530 °C by scanning transmission electron microscopy in high-angle annular dark-field imaging mode (STEM-HAADF). From the image sequences, they identified that the local necking started at a location along the length of the AgNW after 30 s heat-treatment, and eventually the breakup occurred after 120 s.

Previous simulation- [37] and experiment-based [29,34,36] studies have also reported that AgNWs break into more discontinuous segments when increasing the heat-treatment temperature. Wang et al. [36] investigated the effect of heating temperature on a AgNW film where AgNWs with diameters of 20–30 nm and lengths of 25–30 μm are on a polyethylene terephthalate film. Morphologically, their AgNWs were nearly identical to what we used. The AgNW films were heated at 25–250 °C for 20 min by using a digital hotplate. Wang et al. [36] reported that there were no significant morphological changes by heat treatment of 25–150 °C; however, the AgNWs were gradually fused and then split into more and more discontinuous segments as the heat treatment temperature increased from 170 to 250 °C. The morphological changes appeared at lower heat treatment temperatures than our study; 170 and 200 °C heating results of Wang et al. [36] are consistent with our 200 °C (not exhibited in this paper) and 280 °C results, respectively. This may be because the heat treatment method used by Wang et al. (hot plate) [36] was different from ours (oven). 

### 3.2. Oxidation State, Optical Characteristics, and Sheet Resistance

As shown in Figure 2a, the measured Ag 3d core-level XPS spectra (black lines) were fitted with an optimal mixed Gaussian–Lorentzian function. The spectra were split into two separate peaks assigned to Ag (green lines) and silver (Ⅰ) oxide (Ag_2_O, pink lines) [42]. From the XPS-based quantification analysis, the Ag_2_O composition of the AgNW, AgNW-230 °C, and AgNW-280 °C corresponds to 12.9%, 18.2%, and 25.1%, respectively. That is, with increasing the heat-treatment temperature, the oxidation (or Ag-to-Ag_2_O chemical conversion) of the AgNWs proceeded faster. Ag NMs are typically susceptible to chemical conversion into relatively stable Ag_2_O [43,44]. Ag_2_O is a more thermodynamically stable compound than other silver oxide compounds, such as Ag_2_O_3_, Ag_3_O_4_ and Ag_4_O_3_ [44]. Li et al. [45] and Dannenberg et al. [46] examined the generation and growth of Ag_2_O grains over silver materials as an oxidation-induced phenomenon. Li et al. [45] inspected the generation process of Ag_2_O grains through in situ high-resolution transmission electron microscopy (HR-TEM) observation. An AgNP was readily oxidized, resulting in the nucleation and growth of Ag_2_O grains over the surface. The Ag_2_O grains in the form of truncated spherical shells grew up to 10 nm in size. Dannenberg et al. [46] also observed Ag_2_O grain growth up to 40–50 nm over a 80 nm-thick Ag film. They found that the grain growth was limited by their detachment from the Ag film. Ag_2_O grains are likely to fall off from Ag materials since metal-ceramic bonding is generally a weak physical attachment [47]. Therefore, the NPs smaller than 50 nm, detectable in Figure 1c, are speculated as Ag_2_O grains detached from the AgNWs, which is well matched with the curve fitting results of the XPS spectra (Figure 2a). 

Due to the morphological transformation and oxidation of the AgNW film by heat-treatment, the optical properties and sheet resistance changed. Figure 2b shows that the transmittance over the visible-light wavelength region (400–700 nm) slightly decreased with increasing the heat-treatment temperature, which is consistent with the reduced total light transmittance from 91.5% to 89.4% of Figure 2c. This is due to the increased light scattering, as identified from the increased haze from 0.4% to 1.4% in Figure 2c. Incident light is reflected back all when hitting a smooth surface (e.g., mirror), while it is reflected in various directions (i.e., the light-scattering occurs) in the case of a rough surface [48]. Even microscopic irregularities build roughness and induce the scattering of the incident light [49]. Hence, the AgNWs segmentation (breakup) and Ag_2_O NPs generation can increase the haze, reducing the light transmittance. In addition, as another consequence of the segmentation and oxidation, the sheet resistance of the AgNW-230 °C and AgNW-280 °C notably increased from 156 to over 10000 ohm/square. 

### 3.3. Antibacterial Activity

By counting the viable bacteria colonies on incubated agar plates (Figure 3a) in triplicate, the antibacterial activity of the AgNW films in percentage reduction and log reduction was obtained, as shown in Figure 3b,c. In the event of 30 min contact with *S. aureus*, the antibacterial activity of the AgNW, AgNW-230 °C, and AgNW-280 °C was 31.7% ± 2.4% (log reduction of 1.50 ± 0.38), 92.5% ± 2.2% (1.97 ± 0.34), and 94.7% ± 1.7% (1.98 ± 0.32), respectively. For *E. coli*, their antibacterial activity was 57.0% ± 3.0% (1.76 ± 0.48), 84.1% ± 4.9% (1.92 ± 0.69), and 98.7% ± 0.5% (1.99), respectively. The antibacterial effect of the AgNW film increased with increasing the heat-treatment temperature. From an immunological point of view, these increases in bacterial reduction are perceived as meaningful because the number and concentration of bacteria generally plays a decisive role in infection possibility [50,51]. In the case of *S. aureus*, the heated AgNW films did not show improvement in antibacterial activity, as the figures below. Gram-negative bacteria (e.g., *E. coli*) typically possess greater resistance to antibiotics and antibacterial materials than Gram-positive bacteria (e.g., *S. aureus*). Hence, it is suspected that the bare and heated AgNW films could not show any difference in antibacterial activity against weak Gram-positive bacteria (i.e., *S. aureus*) for a short contact time of 30 min. 

It is well known that Ag or Ag-based nanomaterials elicit an antibacterial effect [18,19,20,21,22]. Their antibacterial actions are not yet fully clarified, but it is broadly recognized that one of the main sources of the antibacterial activity is Ag^+^ released from metallic Ag [21,22,23,24,25]. In particular, if the size of Ag-based materials is greater than 10 nm, the Ag^+^ source becomes more important [52,53]. Ivask et al. [54] and Jose et al. [53] found that the most effective source of the antibacterial actions of Ag NPs is determined their size; the antibacterial activity is determined by the released Ag^+^ for the Ag NPs larger than 10 nm in size, whereas the interaction of Ag NPs with the bacterial cell wall is crucial for the Ag NPs less than 10 nm. In the present investigations, all the AgNW films contain AgNMs (e.g., nano-scale wires, droplets, particles) of sizes larger than 10 nm, and hence it is supposed that the AgNW films presented strong antibacterial activity majorly by released Ag^+^. It is supposed that the 2–3-nm thick PVP capping layer of the AgNWs did not meaningfully affect the Ag^+^ release rate and hence antibacterial activity of the AgNW samples. PVP, as a water-soluble polymer, is readily soluble, even in cold water [53], and so the thin capping layer can be promptly dissolved in the used water-based inoculum liquid at the given incubation temperature of 35 °C. As depicted in Figure 4, released Ag^+^ reacts freely with bacterial cells, causing the cells to die by the following antibacterial-action mechanisms: Ag^+^ directly destroys the cell wall consisting of an outer membrane and a single-layered peptidoglycan, and eventually causes the cytoplasmic leakage of cellular contents [55]. Ag^+^ promotes the formation of intracellular reactive oxygen species (ROSs). ROSs, involving superoxide-radical (O_2_^−^), hydroxyl radical (•OH), hydrogen peroxide (H_2_O_2_), and singlet oxygen (^1^O_2_), are short-lived strong oxidants [56]. Ag^+^ mainly generates O_2_^−^ by two mechanisms: one is the inactivation of respiratory chain enzymes by interacting between Ag^+^ and the thiol group of the enzymes [56,57,58], and the other is Ag^+^-mediated mitochondria dysfunction [59,60]. The ROS-induced oxidative stress attacks the ribosome and DNA of bacterial cells, and furthermore oxidizes the membrane lipid excessively [61]. Ag^+^ itself inhibits bacterial reproduction by losing the replication ability of DNA by binding to the DNA [56]. Also, the Ag^+^ denaturizes and impairs the ribosome, leading to the protein-synthesis inhibition and eventually plasmatic membrane degradation [62]. Ag^+^ with high affinity for thiol groups favors a reaction with cysteine residues of respiration chain proteins [63], in a manner that strongly binds to the cysteine residues. As a consequence of the reaction, energy generation (i.e., ATP synthesis) is inhibited due to a decrease in the proton motive force by proton leakage into the cytoplasm [64]. 


In order for Ag^+^ to be released, Ag or Ag-based nanomaterials must primarily go through a process known as oxidative dissolution [65], since Ag is very stable and hence poorly soluble in water [66]. In contrast with Ag, Ag_2_O can be immediately dissolved into the water-based inoculum liquid without further oxidation. With regard to the solubility in water, Ag_2_O (0.025 g/L at room temperature) is much greater than Ag (almost insoluble) [66]. Thus, the heating-induced pre-oxidation (or phase conversion from Ag to Ag_2_O) of the AgNW film promotes the dissolution and so the Ag^+^ release. Moreover, the heating-induced segmentation synergistically increases the solubility [67] and so antibacterial activity [21,22,67]. Furthermore, it is well known that nanoscale protrusions like the generated Ag_2_O NPs (Figure 1c) can physically kill bacteria by causing cytoplasmic leakage by piercing the bacterial cell wall [68,69]. Hence, in this study, the segmentation (Figure 1) and pre-oxidation (Figure 2a) of the AgNW film by heat-treatment is estimated to enhance the antibacterial activity by facilitating the dissolution and activating the physical contact killing mechanism by nano-level Ag_2_O NPs. 

Likewise, the post-heat treatment used in this paper is an effective and easy strategy for improving the antibacterial activity of AgNW films. The bottom-up approach is the most common method to prepare AgNPs with a high antibacterial activity. However, the AgNPs produced by bottom-up synthesis methods (e.g., sol-gel method) are typically prone to agglomeration-induced antibacterial and optical degradation when in a film. On the other hand, the top-down method we used can address the problem thanks to the inherent anti-agglomeration of AgNWs. By applying heat-treatment to an AgNW film, the antibacterial performance of the AgNW film can be maximized by proper segmentation and pre-oxidation without the agglomeration. The anti-agglomeration of as-fabricated and heated AgNW films also makes them transparent. Thus, AgNW films heat-treated at high temperatures are estimated to be promising antibacterial overcoating films that can be applicable to various heat-insensitive stuffs (e.g., metal and ceramic-based household goods, biomedical tools, art pieces, and window) thanks to the potent antibacterial activity and transparency. 

## 4. Conclusions 

In this study, we present a facile technique that effectively enhances the antibacterial activity of AgNW transparent film through the manipulation of a heat-treatment. AgNW films were spin-coated on a glass, and then heated at different temperatures for a period of time in an air oven. As the heat-treatment temperature increased, the AgNWs of the film broke into more discontinuous segments due to the Rayleigh–Plateau instability in the AgNWs. Furthermore, the high heat-treatment temperature accelerated the oxidation. These segmentation and pre-oxidation processes via heat-treatment improved the antibacterial activity against *S. aureus* and *E. coli*; meanwhile, the transparency was slightly reduced. The enhancement in antibacterial activity is because the morphological and chemical changes facilitate the dissolution of the AgNW film, and hence boost the Ag^+^-associated antibacterial actions. Moreover, the generated Ag_2_O NPs synergistically improved the antibacterial activity by activating the contact-killing biocidal mechanism. These findings indicate that a heat-treated AgNW film as an overcoating can confer strong bacterial resistance to various heat-resistant applications (e.g., metal and ceramic-based biomedical and optical applications, and art pieces) without notable optical degradation. 

## Figures and Tables

**Figure 1 nanomaterials-10-00938-f001:**
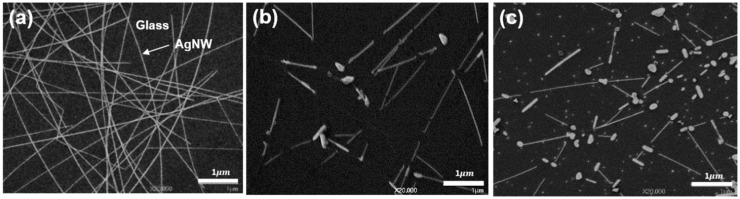
Field emission scanning electron microscopy (FE-SEM) top-view images of (**a**) silver nanowire film (AgNW), (**b**) heated at 230 °C (AgNW-230 °C), and (**c**) 280 °C (AgNW-280 °C).

**Figure 2 nanomaterials-10-00938-f002:**
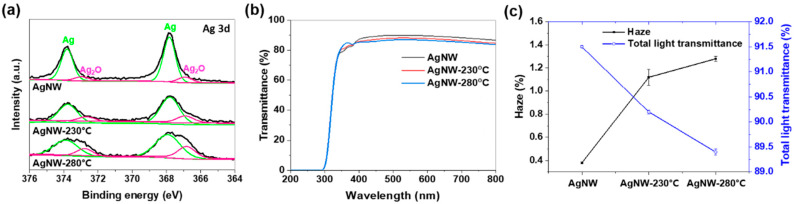
(**a**) Ag 3d core-level XPS high-resolution spectra, (**b**) transmittance and photo, and (**c**) haze and total light transmittance of AgNW, AgNW-230 °C, and AgNW-280 °C.

**Figure 3 nanomaterials-10-00938-f003:**
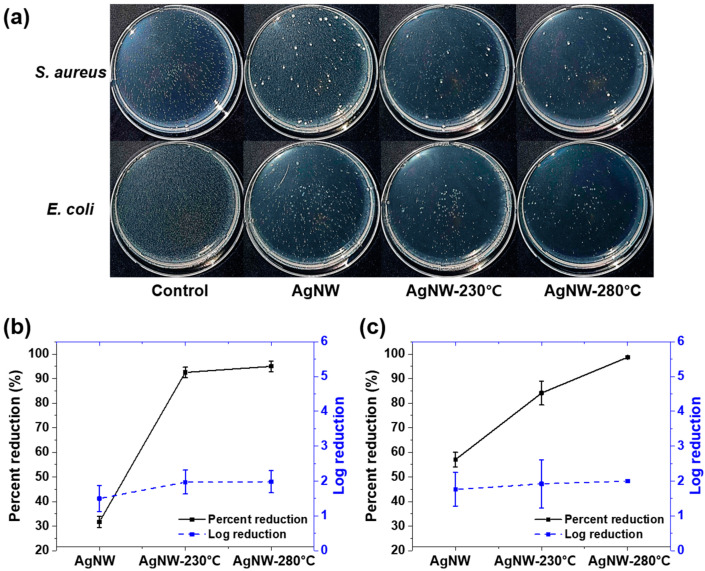
(**a**) Top-view photos of incubated agar-plates and antibacterial activity of AgNW, AgNW-230 °C, and AgNW-280 °C against (**b**) *S. aureus* and (**c**) *E. coli.* The bacterial tests were performed by contacting the sample with an inoculum containing 4 × 10^6^ CFU of test bacterial strains for 30 min.

**Figure 4 nanomaterials-10-00938-f004:**
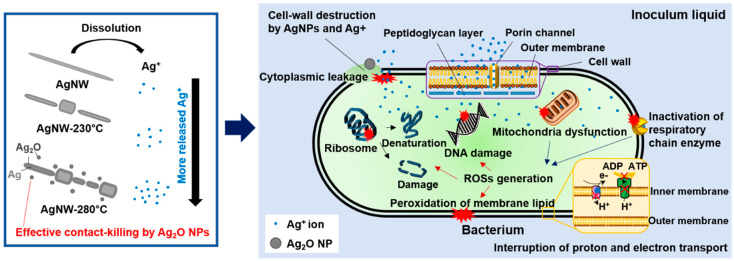
Schematic illustration of the antibacterial mechanisms of AgNW, AgNW-230 °C, and AgNW-280 °C.

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
