# Peer review of "Enhancement of Antibacterial Performance of Silver Nanowire Transparent Film by Post-Heat Treatment"

_nanomaterials, 2020, doi:10.3390/nano10050938_

Round 1

Reviewer 1 Report

The authors address an interesting aspect of the effect of heat treatment of Ag nanowires. Although this feature might be of interest also for optical and electronic devices based on AgNWs, the authors focus here on the antibacterial activity oi heat-treated AgNWs.

The results are well presented but limited only to basically two samples, differing in the heating temperature. The paper might be published although the piece of information provided is limited, additional comments and possibly experiments should be added.

For example:

  1. the authors conclude that the proposed heating treatment could be used for the coating of "household goods, biomedical tools, art pieces, and window", implying that the heat-treatment should be run before coating. On the other hand, the results only describe the heating of spin-coated samples, this cast some confusion. Did the authors try also to spin-coat pre-heated AgNw dispersions? Were the results comparable?
  2. Did the authors try alternative coating methods (Drop-casting or Dip-coating)? 
  3. The authors focus only on two temperatures, did they also perform other experiments to find a trend or a threshold? Such results could be an important piece of information.
  4. Antibacterial activity was studied only for E.coli, were other bacteria tested? the results sometimes may differ-
  5. Concerning the percentage reduction, an increase from 92.6 to 95.7 % is really relevant? The authors could comment and add appropriate references. 
  6. Figure 2 and figure 3. The quality of the photos is in general poor, maybe a change in the background color could help. The photographs in picture 2 should be discussed.
  7. Figure 2. The decrease in transmittance is indeed small, it is hard to appreciate the authors' statement (page 5, line 174'175): "Due to the morphological transformation and oxidation of the AgNW film by heat-treatment, the optical properties, and sheet resistance changed".
  8. On the other hand, the change in sheet resistance is remarkable, this finding could be stressed to caution scientists using AgNWs to enhance conductivity to avoid heat-treatment. Were there differences in sheet resistance for the two heat'treatments (230 °C and 280 °C )?  

Author Response

We thank you for your critical and careful review of our work and your constructive suggestions and comments, which have helped us to substantially improve the manuscript. We included the page and line numbers of the changes in the unmarked revised manuscript. In the following, we address your concerns point by point.

Reviewer 2 Report

Major comments

1. The authors have only test one strain of E. coli. To confirm the improvement of the antibacterial activity of the AgNW, they need to test first other MDR clinical isolates of Escherichia. coli and other Gram-negative bacilli such as Klebsiella pneumonia, Pseudomonas aeruginosa and Acinetobacter baumannii.

2. It is no clear how they calculate the % of reduction of bacterial growth in the presence of AgNW at different conditions.

Minor comments

1. The name and source of E. coli strain used in this study is missing.

2. Remove the figure 3C from figure and move it to a new figure.

Author Response

Thank you for taking the time to review our original manuscript and to provide your careful and professional feedback. have included the page and line numbers of the changes made in the unmarked revised manuscript. In the following, we address your concerns point by point.

Round 2

Reviewer 1 Report

Although the authors corrected quickly the manuscript eliminating verbose paragraphs summarizing established literature and removing useless fotographs, I do not agree with them on the following points:

Reviewer Point 2: Did the authors try alternative coating methods (Drop-casting or Dip-coating)?

Authors reply: We needed to fabricate a very thin AgNW film for high transparency, so the spincoating method operating at a high rotation speed (i.e., 7000 rotations per minute) was selected. Based on our experience, we supposed that other methods are not suitable for making such thin and uniform thickness unless the AgNW dispersion is diluted.

I argue that either their experience is limited or they do not want to bother to run control experiments with these methods. A fast check on drop-casted or dip-coated samples would confirm their point or enlarge the preparation possibilities.

Accurate drop-casting works indeed very well to prepare uniform thin films either as AgNw monolayer or superimposed monolayers- Langmuir'Blodgett fabrication would ensure even higher uniformity imparting parallel orientation of the nanowires but is indeed a laborious procedure.

In any case, a comment (either negative or positive based on their experience) on the possible use of different preparation methods as an alternative to spin-coating should be added in the text.

Reviewer Point 3: The authors focus only on two temperatures, did they also perform other experiments to find a trend or a threshold? Such results could be an important piece of information.

Authors reply: We selected heating temperatures by considering the morphology of the AgNW segments (Figure 1). Up to 200 °C, the morphology of the AgNWs was sustained. On the other hand, the AgNWs generally thickened from 280 °C. Theoretically, this overall thickening decreases the specific surface area and hence antibacterial activity. So, we figured out the heating temperature effect on the morphology; heating temperatures of 200–300 °C (at 10 °C intervals) were assessed. By considering the morphology of Ag segments, we selected two representative temperatures, 230 °C and 280 °C

I understand the author point reported in this sentence "There were no significant morphological changes by heat treatment of 25–150 °C, however the AgNWs were gradually fused and then split into more and more discontinuous segments as the heat treatment temperature increased from 170 to 250 °C [36]".

Nevertheless, the addition of an experiment at T=200 °C and 250 °C in figures 2c and 3b would be a piece of crucial information allowing for the choice of the treatment temperature and avoiding unnecessary over-heating in future studies.

Reviewer Point 4: Antibacterial activity was studied only for E.coli, were other bacteria tested? The results sometimes may differ

Authors reply: We performed antibacterial tests against E. coli and S. aureus which are representative Gram-negative bacteria and Gram-positive bacteria. In the case of S. aureus, the heated AgNW films didn’t show improvement in antibacterial activity, as the figures below. Gram-negative bacteria (e.g., E. coli) typically possess greater resistance to antibiotics and antibacterial materials than Gram-positive bacteria (e.g., S. aureus). Hence, it is suspected that the bare and heated AgNW films couldn’t show any difference in antibacterial activity against weak Gram-positive bacteria (i.e., S. aureus) for a short contact time of 30 min.

The addition of this paragraph on the control experiment in the text would be useful. Did the authors, by chance also use another gram-negative bacteria Pseudomonas Aeruginosa? This bacteria may be inhibited by nanowires (but not as efficiently by nanospheres) and it would be interesting to see the effect of heat-treated nanowires.

Author Response

Thank you very much for your comments and suggestions, which have helped us to greatly improve the manuscript. We included the page and line numbers of the changes in the unmarked revised manuscript. In the following, we address your concerns point by point.

Reviewer 2 Report

The authors have responded to my comments.

Author Response

Thank you very much for your comments and suggestions, which have helped us to greatly improve the manuscript.